# Maternal Dietary Deficiency in Choline Reduced Levels of MMP-2 Levels in Blood and Brain Tissue of Male Offspring Mice

**DOI:** 10.3390/cells13171472

**Published:** 2024-09-02

**Authors:** Mitra Esfandiarei, Shawn G. U. Strash, Amanda Covaleski, Sharadyn Ille, Weidang Li, Nafisa M. Jadavji

**Affiliations:** 1Department of Biomedical Sciences, Midwestern University, Glendale, AZ 85308, USA; mesfan@midwestern.edu; 2Anesthesiology, Pharmacology, & Therapeutics, Faculty of Medicine, University of British Columbia, Vancouver, BC V6T 1Z4, Canada; 3Basic Medical Sciences, College of Medicine Phoenix, University of Arizona, Phoenix, AZ 85004, USA; 4College of Osteopathic Medicine, Midwestern University, Glendale, AZ 85308, USA; sstrash@asu.edu; 5College of Pharmacy, Midwestern University, Glendale, AZ 85038, USA; amanda.covaleski@midwestern.edu; 6College of Dental Medicine, Midwestern University, Glendale, AZ 85038, USA; sharadyn.ille@midwestern.edu; 7College of Veterinary Medicine, Midwestern University, Glendale, AZ 85038, USA; wli@midwestern.edu; 8Department of Biomedical Sciences, Southern Illinois University, Carbondale, IL 62901, USA; 9Department of Child Health, College of Medicine Phoenix, University of Arizona, Phoenix, AZ 85721, USA; 10Department of Neuroscience, Carleton University, Ottawa, ON K1S 5B6, Canada

**Keywords:** maternal nutrition, folic acid, choline, ischemic stroke, offspring, one-carbon metabolism

## Abstract

Ischemic stroke is one of the leading causes of disability and death globally, with a rising incidence in younger age groups. It is well known that maternal diet during pregnancy and lactation is vital for the early neurodevelopment of offspring. One-carbon (1C) metabolism, including folic acid and choline, plays a vital role in closure of the neural tube in utero. However, the impact of maternal dietary deficiencies in 1C on offspring neurological function following ischemic stroke later in life remains undefined. The aim of this study was to investigate inflammation in the blood and brain tissue of offspring from mothers deficient in dietary folic acid or choline. Female mice were maintained on either a control or deficient diet prior to and during pregnancy and lactation. When offspring were 3 months of age, ischemic stroke was induced. One and a half months later, blood and brain tissue were collected. We measured levels of matrix metalloproteases (MMP)-2 and 9 in both plasma and brain tissue, and reported reduced levels of MMP-2 in ChDD male offspring in both tissue types. No changes were observed in MMP-9. This observation supports our working hypothesis that maternal dietary deficiencies in folic acid or choline during early neurodevelopment impact the levels of inflammation in offspring after ischemic stroke.

## 1. Introduction

Ischemic stroke is a leading cause of long-term neurological impairments worldwide, often resulting in disability [1,2,3]. An ischemic stroke occurs when there is a blockage in a blood vessel in the brain; an example is a thrombus. The factors that increase risk for ischemic stroke are on the rise; these include hypertension, obesity, and diabetes [4,5,6]. Nutrition is linked to the onset of these risk factors [7] and, interestingly, is a modifiable risk factor for ischemic stroke [8,9]. Previous research in our laboratory has established a link between one-carbon (1C) metabolism deficiency and worse ischemic stroke outcome [10,11,12,13]. Folates and choline are the main components of 1C and both are well known for their role in the closure of the neural tube of the developing fetus during pregnancy [14].

Folic acid is the synthetic form of a B-vitamin and is also more bioavailable [15]. Folic acid is involved in several cellular processes. Once it is transported into the cell by the folate receptor, folic acid is converted to tetrahydrofolate which then aids in the production of purines. Through thymidylate synthase, folic acid is also involved in DNA repair, specifically the removal of uracil from DNA. Lastly, once folic acid has been reduced by methylenetetrahydrofolate reductase, it is converted to the main circulating form of folic acid, 5-methylTHF. This form of folic acid aids in metabolizing homocysteine through a methylation reaction with the assistance of vitamin B12. Elevated levels of homocysteine are well associated with increased risk in cardiovascular diseases, like stroke [16]. Choline is a nutrient that plays an important role in lipid and homocysteine metabolism. It also plays an essential role in the synthesis of acetylcholine, an important neurotransmitter in the nervous system. Within the 1C metabolic pathway, choline generates methyl groups to reduce levels of homocysteine; this is achieved through betaine and betaine homocysteine methyltransferase. Our previous work has shown that when animals are deficient in dietary folic acid, choline is used [17].

During pregnancy, there is an increased demand for maternal dietary intake of folic acid and choline [18]. As described above, folic acid plays an important role in nucleotide synthesis and DNA repair. Both folic acid and choline are involved in methylation through their involvement in 1C [19]. These cellular processes are pivotal during pregnancy and important during normal neurodevelopment. The brain is an organ that takes over 20 years to develop; in utero, both folic acid and choline play an important role in closing the neural tube, which is the developing brain and spinal cord. Less is understood about the impact of maternal dietary deficiencies of folic acid or choline on offspring neurological function after birth.

The Developmental Origins of Health and Disease (DOHaD) theory suggests that prospective chronic diseases are programmed in utero [20]. In offspring, maternal dietary deficiencies have been associated with modified neural tube closure [21,22] and neurocognitive development [23,24]. Epidemiological studies have demonstrated the effect of the maternal diet on lifelong cardiovascular and neurological function [25]. Female and male offspring from moms deficient in folic acid or choline have a worse outcome after ischemic stroke, including impaired behavior in both 4.5- and 11.5-month-old animals [26,27]. We also report reduced blood flow in female offspring [28]. In studies investigating mechanisms, we reported reduced levels of apoptosis, response to hypoxia, and neuroinflammation when measured 1.5 months after ischemic damage was induced. In this study, we wanted to investigate other mechanisms, such as inflammation, at the same timepoint.

Matrix metalloproteases (MMPs) have been identified as proteins that become concentrated in areas of tissue undergoing response to injury and acute inflammation [29,30]. MMPs have been correlated with a variety of diseases related to chronic inflammation or neurological disorders, such as ischemic stroke. Following injury, the increased expression of MMPs has been related to increased blood–brain barrier (BBB) permeability due to the disruption of the tight junctions and adherens junctions in the BBB. This increased permeability results in the further exacerbation of potential CNS damage [9,31]. It is reported that mice with reduced production of MMP-9 (a type of gelatinase MMP, along with MMP-2) following stroke presented with less cerebral injury compared to control mice [32]. A separate study also found that mice with increased production of MMP-9 had developed exacerbated cerebral injury compared to control groups [33]. These findings suggest the possibility of MMPs being a potential therapeutic target for ischemic injuries.

Maternal dietary deficiencies in folic acid or choline impact offspring stroke outcome in female and male offspring [26,27,28]; however, the mechanism through which this occurs is not well understood. The aim of this study was to measure the levels of the inflammatory markers MMP-2 and -9 post-stroke in the brain and plasma of female and male offspring from mothers deficient in dietary folic acid or choline during pregnancy and lactation.

## 2. Materials and Methods

All experiments were conducted in accordance with the guidelines of the *National Institutes of Health Guide for the Care and Use of Laboratory Animals* (8th ed., 2011) and the University Institutional Animal Care Users Committee (IACUC 2983) in February 2020. Female and male C57/BL6J (RRID: IMSR_JAX:000664) mice were obtained from Jackson Laboratories (Bar Harbor, ME, USA). All mice were housed in a controlled environment with a 12 h light/dark cycle, a temperature of 22–23 °C, a relative humidity of 55% to 60%, and free access to food and water. Experimental male and female offspring were generated from breeding pairs.

The experimental manipulations are summarized in Figure 1. Two-month-old female mice were habituated for 7 days before they were fed on either a control (CD, Envigo, Indianapolis, IN, USA, Catalog: # TD.190790), folic acid (FADD, Envigo, Indianapolis, IN, USA, Catalog: # TD.01546), or choline-deficient diet (ChDD, Envigo, Indianapolis, IN, USA, Catalog: # TD.06119). All diets contained 1% succinylsulfathiazole to prevent folate synthesis by intestinal flora. The CD represented adequate levels of both folic acid and choline that would be found in a conventional diet, which were determined from the previous literature and experimentation [34,35,36]. The deficient diets consisted of lower levels of folic acid or choline compared to the control diet. The CD contained 2 mg/kg of folic acid and 1150 mg/kg of choline bitrate, whereas the FADD contained only 0.3 mg/kg and the ChDD only had 300 mg/kg of choline bitrate. The dams were maintained on the diets 4 weeks prior to pregnancy, and during pregnancy and lactation. Once female and male offspring were weaned from their mothers, they were maintained on a CD for the duration of the experiment. At 2 months of age, the offspring were subjected to ischemic stroke, using the photothrombosis model on the sensorimotor cortex. Four and a half weeks after damage, blood and brain tissue were collected for further analysis. For each dietary group, we had 6 female and male breeding pairs set up. The following numbers are the numbers of offspring generated: CD (female: 15, male: 13), ChDD (female: 13, male: 14), and FADD (female: 15, male: 12).

### 2.1. Photothrombosis

At 2 months of age, all female and male offspring were subjected to photothrombosis to induce a unilateral ischemic stroke in the sensorimotor cortex. They were anesthetized with isoflurane (Henry Schein, Melville, NY, USA, Catalog: #NDC-11695-6776-1) (1.5%) in a 70:30 nitrous oxide/oxygen mixture. Core body temperature was monitored with a rectal thermometer (Harvard Apparatus, Holliston, MA, USA) and maintained at 37 ± 0.2 °C using a heating blanket. The photosensitive Rose Bengal dye (10 mg/kg; (Millipore Sigma, Burlington, MA, USA, Catalog: #198250-25G) was injected intraperitoneally 5 min prior to irradiation. A 532 nm green laser (Beta Electronics, Irvine, CA, USA (#MGM20 (20–25 mW)) was placed 3 cm above the animal and directed to the sensorimotor cortex (mediolateral + 0.24 mm) [10,11,26,27,28] for 15 min. All animals received 0.1 mg/kg of Buprenorphine HCl after damage to assist with post-operative pain. After the completion of surgery, the animals were singly housed. Animals were provided with mash and hydrogel for up to one week post-stroke and housed individually.

### 2.2. Brain Tissue Collection, Processing, and Sectioning

At 4.5 weeks after stroke, female and male offspring mice were anesthetized with an overdose of CO_2_. Brain tissue was dissected from the skull and placed into 4% paraformaldehyde overnight. The solution was replaced with 10% sucrose and then 20% sucrose, and each change in solution lasted overnight. Once the brain tissue was ready to be sectioned, it was frozen and mounted. Then, the frozen brain tissue was sectioned using a Thermo HM550 cryostat (Fisher Scientific, Hampton, NH, USA) at a thickness of 30 µm and the sections were slide-mounted on microscope slides in serial order. The microscope slides were stored at −80 °C until analysis.

### 2.3. Immunofluorescence 

Brain tissue was used for immunofluorescence analysis to assess inflammation, particularly levels of MMP-2 and 9. The primary antibodies used included MMP-2 (1:100, AbCam, Cambridge, MA, USA, Catalog: #ab97779, RRID: AB_10696122) and MMP-9 (1:500, AbCam, Cambridge, MA, USA, Catalog: #ab228402, RRID: AB_2910609). All brain sections were stained with a marker for neuronal nuclei (NeuN, 1:200, AbCam, Cambridge, USA, Catalog: # ab104224, RRID: AB_10711040). Primary antibodies were diluted in 0.5% Triton X diluted in phosphate-buffered saline and incubated with brain tissue overnight at 4 °C. Three rinses were conducted between each step using phosphate-buffered saline; each rinse was 5 min in length. The next day, brain sections were incubated with the secondary antibodies Alexa Fluor 488 and 555 (Cell Signaling Technologies, Danvers, MA, USA, Catalog: # 4408 and 4413) at room temperature for 2 h. Brain tissue was also stained with 4′,6-diamidino-2-phenylindole (DAPI) (1:10,000, Thermo Fisher Scientific, Waltham, MA, USA, Catalog: # EN62248). Tissue sections were cover-slipped with Fluromount and stored at 4 °C in the dark until visualization. The staining was visualized using an Revlove microscope (Echo, San Diego, CA, USA) and all images were collected at the magnification of 200×.

In brain tissue within the ischemic core region, the co-localization of MMP-2 or MMP-9 with NeuN-labeled neurons was counted and averaged per animal. Images were merged, and a positive cell was indicated by the co-localization of the antibodies of interest located within a defined cell. Cells were distinguished from debris by identifying a clear cell shape and intact nuclei (indicated by DAPI and NeuN) under the microscope. All cell counts were conducted by three individuals blinded to the treatment groups. Using ImageJ, the number of positive cells were counted in three brain sections per animal; a total of 3 to 4 animals were analyzed per group. For each section, three brain sections were analyzed per animal. Within each brain section, three subsections were measured. The number of positive cells was averaged for each animal.

### 2.4. ELISA

Blood was collected in EDTA-coated tubes from experimental animals at the point of euthanasia using cardiac puncture. Samples were spun down at 7000 G for 7 min at 4 °C and plasma was stored at −80 °C until analysis. The mouse MMP-2 ELISA Kit (Abcam Inc., Cambridge, MA, USA, Catalog: #ab254516) and the MMP-9 ELISA Kit (Abcam Inc., Cambridge, MA, USA, Catalog: #ab253227) were used for the quantitative measurement of MMP-2 and MMP-9 in mouse plasma, respectively, according to the manufacturer’s instructions. In brief, the plasma samples were diluted according to the recommended dilution factor provided in the kit protocol, and 50 µL from each sample was added to each well. Then, a 50 µL antibody cocktail was added to each well. A substrate solution of 3,3′,5,5′-Tetramethylbensidine (TMB) was applied for visualization. The absorbance at 450 nm was measured using a BIOTEK EPOCH (Agilen, Sanata Clara, CA, USA) microplate reader.

### 2.5. Data Analysis and Statistics

GraphPad Prism 10.0 was used to analyze cell counts and ELISA data. In GraphPad Prism, the D’Agostino–Pearson normality test was performed prior to two-way ANOVA analysis for all data. The two-way ANOVA compared the mean measurement of both sex and maternal dietary groups for ELISA and immunofluorescence staining. One-way ANOVA analysis was used to compare maternal dietary differences when there were no offspring sex differences. The significant main effects of two- and one-way ANOVAs were followed up with Tukey’s post hoc test to adjust for multiple comparisons. All data are presented as mean ± standard error of the mean (SEM). Statistical tests were performed using a significant *p* value of 0.05.

## 3. Results

### 3.1. Plasma Levels of MMP-9 and MMP-2

Four and a half weeks after ischemic stroke, we measured the levels of MMP-2 and MMP-9 in plasma isolated from female and male offspring from mothers maintained on the CD, ChDD, or FADD prior to and during pregnancy and lactation. Offspring MMP-2 levels were affected by maternal diet (Figure 2A, F(_2,73_) = 5.82, *p* = 0.007). Male ChDD mice had lower levels of MMP-2 compared to FADD mice (*p* = 0.024). MMP-2 levels were not impacted by sex (F(_1,73_) = 2.20, *p* = 0.13), nor was there an interaction between sex and maternal diet (F(_2,73_) = 1.19, *p* = 0.031). The maternal diet offspring were exposed to in utero and during lactation did not impact MMP-9 levels (Figure 2B; F(_2,73_) = 0.14, *p* = 0.867), sex (F(_1,73_) = 0.97, *p* = 0.97), or the interaction between maternal diet and sex (F(_2,73_) = 0.82, *p* = 0.45).

### 3.2. Brain Levels of MMP-2 and MMP-9

To assess inflammation in the brain tissue, we measured the neuronal levels of MMP-2 and MMP-9 within the ischemic damage region. Representative images of MMP-2 and NeuN staining are shown in Figure 3A. The tissue was collected four and a half weeks after ischemic stroke. Maternal diet did impact MMP-2 levels (Figure 3B, (F(_2,14_) = 5.00, *p* = 0.02). There were no significant pairwise comparison differences. Furthermore, there was no impact of offspring sex (F(_1,14_) = 0.82, *p* = 0.95) or the interaction between maternal diet and offspring sex (F(_2,14_) = 1.09, *p* = 0.37).

Since there was no sex difference shown between offspring, using one-way ANOVA, we looked at the impact of maternal diet separately in female and male offspring. There was no impact of maternal diet on the female levels of MMP-2 in brain tissue (F(_2,7_) = 1.183, *p* = 0.361). Interestingly, we observed reduced levels of MMP-2 in male brain tissue because of the maternal diet (F(_2,7_) = 5.734, *p* = 0.034); specifically, between ChDD and CD offspring (*p* = 0.023; Figure 3B).

Representative images of MMP-9 and NeuN staining are shown in Figure 4A. MMP-9 levels were measured in brain tissue within the ischemic region; there was no impact of maternal diet (Figure 4B, F(_2,11_)= 1.33, *p* = 0.30), offspring sex (F(_1,11_)= 0.23, *p* = 0.64), or the interaction between the maternal diet and offspring sex (F(_2,11_)= 0.71, *p* = 0.71).

## 4. Discussion

This study aimed to investigate whether pro-inflammatory MMPs play a role in the impact of reduced maternal dietary levels of folic acid or choline on post-stroke outcomes in female and male offspring. We report that MMP-2 levels were decreased in the brain tissue and plasma of our choline-deficient-diet male offspring following ischemic stroke, whereas MMP-9 levels were unaffected in the treatment groups. Ischemic stroke leads to negative health outcomes around the world [1,2,3]. Nutrition is a risk factor for ischemic stroke [8,9]. The previous literature has identified a link between post-stroke outcome and 1C metabolism [10,11,12,13]. The DOHaD theory suggests that prospective chronic diseases are programmed in utero—giving rise to the programming of offspring cardiovascular, metabolic, and neuroendocrine dysfunctions [20]. Despite impressive evidence of the importance of the maternal environment for fetal growth and development in utero, there is not much information available on the impact on offspring as they age. Nutrition is a modifiable risk factor and can also impact outcome after ischemic stroke [9]. Low dietary levels of maternal folic acid or choline worsen stroke outcomes in female and male offspring [26,27].

MMPs have been established as potential therapeutic targets after ischemic stroke due to their importance and critical role in the inflammation cascade. Decreased production of MMPs following ischemic stroke has already been shown to result in decreased injury to the brain and blood–brain barrier in mice [32,33]. Interestingly, our results show that male offspring from maternal mice with choline-deficient diets had decreased MMP-2 levels present in their plasma and brain tissue following ischemic stroke. It is unexpected that the treatment groups who were exposed to maternal deficiencies of choline or folic acid during in utero and early postnatal development are seemingly able to correct for ischemic stroke more efficiently than their 1C sufficient control group comparison.

MMPs are inflammatory molecules leading to the disruption of barrier-like cellular structures such as tight junctions and adherens junctions, and blood–brain barrier dysfunction. Like other molecules in the inflammatory cascade, these molecules are important in biological processes such as immune support and wound healing, but when these molecules gather in excess, we see the negative consequences of inflammation. MMPs have been shown to be important in protective and beneficial inflammatory responses during normal physiological processes such as bone remodeling and angiogenesis. Dietary deficiencies and their impact on MMP expression and activity in the brain have been studied extensively. Previous studies have shown that deficiencies in vitamin B (B6, B12, and folate) can elevate homocysteine levels, which in turn can increase MMP activity, contributing to neurovascular and brain damage [13,37]. Deficiencies in vitamin E and other antioxidants like selenium can also results in oxidative stress, increasing MMP activity in the brain [34,38].

In our previous reports, we have shown that offspring from moms deficient in folic acid or choline have shown worse outcomes after ischemic stroke, including impaired behavior at both 4.5 and 11.5 months of age [26,27]. We also reported reduced levels of apoptosis, response to hypoxia, and neuroinflammation at 1.5 months after ischemic injury. Hence, the observation of reduced MMP levels in offspring after ischemic stroke might have been expected, since, in the brain, MMP-2 drives neuronal apoptosis and the breakdown of specific substrates such as dystroglycan, a transmembrane receptor involved in the anchoring of astrocyte end feet to the basement membrane via laminin binding. Furthermore, cellular influx into areas of inflammation is regulated by MMP-2 activity [35]. The observed decrease in MMP-2 levels in ChDD mice may be due to the activation of compensatory pathways to counterbalance the deficiency, leading to an adaptive reduction in MMP expression. In addition, maternal protein deficiency could induce epigenetic modifications in the offspring, altering the expression of genes involved in inflammation and MMP regulation. Dietary deficiencies might also affect the development of the offspring’s immune system, resulting in a less robust inflammatory response and consequently lower MMP levels. Another possibility is that the offspring might develop protective adaptations against inflammation and oxidative stress due to chronic maternal deficiency during the developmental stage, leading to reduced MMP levels post-stroke.

From an experimental design perspective, these plasma and brain samples were taken 32 days (4.5 weeks after ischemic stroke) after ischemic stroke was induced. The results could have potentially differed if samples were taken at different points in time, such as sample collection right after ischemic stroke. Potential future studies could include analyzing the same samples across different time frames and assessing if different levels of MMPs were noticed at different points in time. Previous research measured MMP levels at different times following cardiac ischemia. They noticed that MMP levels are increased in proportion to the level of ischemia at the initial onset of ischemia (acute myocardial infarction vs. stable coronary artery disease), but later decrease to below less severe ischemic levels [36]. This could potentially explain the results of our study. The diet-deficient treatment groups may have undergone more severe ischemia than the control groups, having more MMPs initially, but could potentially still have lower levels of MMP a short time later. The previously mentioned study regarding heart ischemia took their samples 3 months post ischemic injury; our results show similar findings at the 1.5-month mark.

## 5. Conclusions

We report reduced levels of MMP-2 in the brain and plasma tissue of male offspring 1.5 months after undergoing induced ischemic stroke from mothers deficient in choline. Previous research has shown that reduced levels of MMPs are associated with less severe injury and complications following cerebral ischemia. However, it would be an oversimplification to classify MMPs as inherently deleterious. MMPs are crucial players during tissue remodeling, repair, and healing processes, but their overexpression and uncontrolled activity can be detrimental, leading to the excessive degradation of extracellular matrix components, contributing to inflammatory and pathological conditions. Further research is necessary to elucidate the precise benefits and adverse consequences of varying levels of MMPs in the brain of offspring from mothers with dietary deficiencies. Additionally, studying MMP expression at different timepoints post-stroke is crucial to understanding their dynamic roles and impacts on neurological recovery and damage. Furthermore, MMP-2 is constitutively expressed in several tissues and is regulated by tumor necrosis factor-α under the influence of the NF-κB transcription factor [35]—investigating these levels in offspring brain tissue is also a future goal. Additionally, inhibiting MMP-2 function may provide some clues to the role of this inflammatory marker on offspring stroke outcome.

## Figures and Tables

**Figure 1 cells-13-01472-f001:**
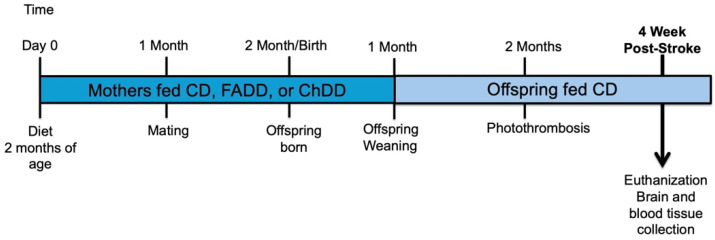
Experimental timeline of study. At 2 months of age, female mice were placed on a control (CD), folic acid (FADD), or choline-deficient (ChDD) diet. Females were maintained on diets for the duration of pregnancy and lactation until the offspring were weaned. Once female and male offspring were weaned, they were maintained on the CD. At 2 months of age, the offspring were subjected to ischemic stroke via a unilateral photothrombosis model. At 3.5 months, female and male offspring were euthanized, and blood and brain tissues were collected for analysis.

**Figure 2 cells-13-01472-f002:**
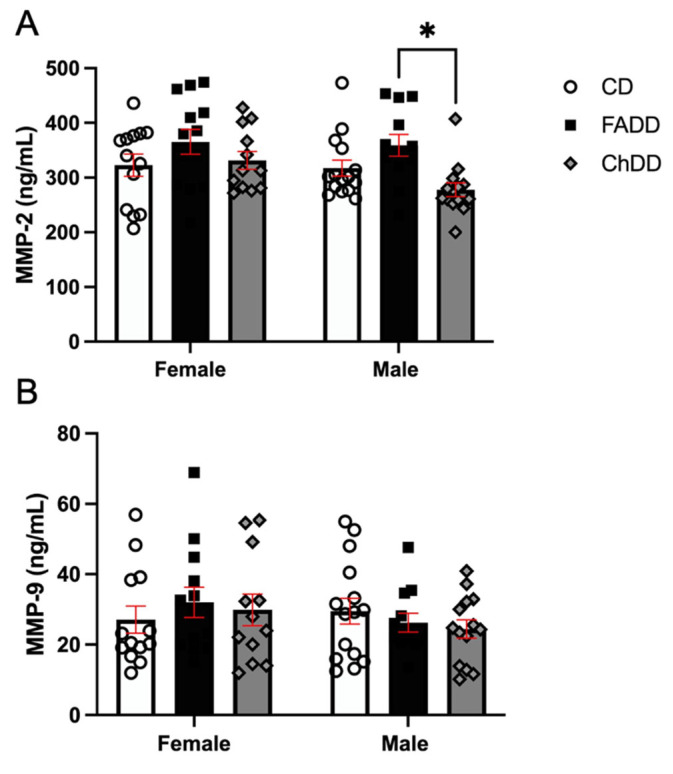
Impact of maternal diets deficient in folic acid (FADD) or choline (ChDD) and control diet (CD) during pregnancy and lactation on levels of matrix metalloproteases (MMPs)-2 (**A**) and 9 (**B**) in plasma of female and male offspring 1 month after ischemic stroke. Mean ± SEM of 10 to 14 animals per group. * *p* < 0.05, Tukey’s pairwise post hoc analysis between male FADD and ChDD offspring.

**Figure 3 cells-13-01472-f003:**
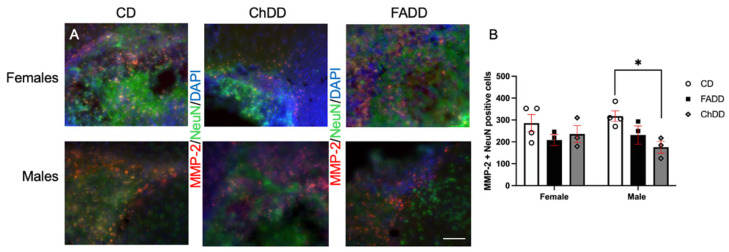
Impact of maternal diets deficient in folic acid (FADD) or choline (ChDD) and the control diet (CD) during pregnancy and lactation on neuronal levels of matrix metalloprotease (MMP)-2 in ischemic brain tissue of female and male offspring one month after ischemic stroke. Representative images of MMP-2 immunofluorescence colocalization with neuronal nuclei (NeuN) and 4′,6-diamidino-2-phenylindole (DAPI) (**A**) and semi-quantitative quantification of MMP-2 levels within ischemic region of brain tissue (**B**). Mean ± SEM of 3 to 4 animals per group. * *p* < 0.05, Tukey’s pairwise post hoc analysis between male CD and ChDD offspring (one-way ANOVA analysis between males and significant maternal diet main effect). Magnification 200X, scale bar 50 µm.

**Figure 4 cells-13-01472-f004:**
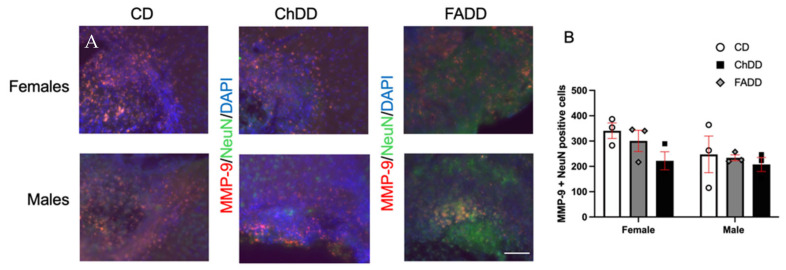
Impact of maternal diets deficient in folic acid (FADD) or choline (ChDD) and the control diet (CD) during pregnancy and lactation on neuronal levels of MMP-9 in the ischemic brain tissue of female and male offspring one month after ischemic stroke. Representative images of matrix metalloprotease (MMP)-9 immunofluorescence colocalization with neuronal nuclei (NeuN) and 4′,6-diamidino-2-phenylindole (DAPI) (**A**) and semi-quantitative quantification of MMP-9 levels within ischemic region of brain tissue (**B**). Mean ± SEM of 3 to 4 animals per group. Magnification 200×, scale bar 50 µm.

## Data Availability

The original contributions presented in the study are included in the article/Appendix A, further inquiries can be directed to the corresponding author.

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
