# Peer review of "Maternal Dietary Deficiency in Choline Reduced Levels of MMP-2 Levels in Blood and Brain Tissue of Male Offspring Mice"

_cells, 2024, doi:10.3390/cells13171472_

Round 1

Reviewer 1 Report

Comments and Suggestions for Authors

Esfandiarei et al investigated here levels of matrix-metalloproteases -2, and -9 in blood and brain tissues of offspring from mothers deficient in dietary folic acid or choline after induced ischemic stroke events.

The MS is well-written, with clear objectives and sufficiently robust methodology to support the main findings and conclusions presented.

However, there are still MINOR observations that should be addressed by the authors before its full acceptance in Cells/MDPI:

(1) Introduction needs a stronger presentation of the metabolic roles of folic acid and choline in C1-metabolism, from my point of view.

(2) I would like to suggest the authors to minimize their conclusions about the supposed inflammation processes, since they actually measured MMP-2, and -9 levels in fluids/tissues. The whole inflammatory scenario should be better described by analyses of many other biomarkers, such as cytokines (TNF-alpha, IL-10, IL-8, etc), as well as immune cell chemotactic compounds. 

Comments on the Quality of English Language

Few sentences were not fully clear to me, but I presume an additional English review would be sufficient to correct the updated version 

Author Response

Reviewer 1

1. Esfandiarei et al investigated here levels of matrix-metalloproteases -2, and -9 in blood and brain tissues of offspring from mothers deficient in dietary folic acid or choline after induced ischemic stroke events.

The MS is well-written, with clear objectives and sufficiently robust methodology to support the main findings and conclusions presented.

Response: We thank the reviewer for their positive comments.

However, there are still MINOR observations that should be addressed by the authors before its full acceptance in Cells/MDPI:

2. Introduction needs a stronger presentation of the metabolic roles of folic acid and choline in C1-metabolism, from my point of view.

Response: We thank the reviewer for this information. We have added more information about folates and choline to the introduction. The text can be found on pages 1 to 2, lines 46 to 60. The revised text has been copied below for reference.

Folic acid is the synthetic form of the B-vitamin and is also more bioavailable[15]. Folic acid is involved in several cellular processes, once it is transported into the cell by the folate receptor, folic acid is converted to tetrahydrofolate which then aids in the production of purines. Through thymidylate synthase, folic acid is also involved in DNA repair, specifically the removal of uracil from DNA. Lastly, once folic acid has been reduced by methylenetetrahydrofolate reductase it is converted to the main circulating form of folic acid, 5-methylTHF. This form of folic acid aids in metabolizing homocysteine through a methylation reaction with the assistance of vitamin B12. Elevated levels of homocysteine are well associated with increased risk in cardiovascular diseases, like stroke [16]. Choline is a nutrient that plays an important role if lipid and homocysteine metabolism. It also plays an essential role in the synthesis of acetylcholine, an important neurotransmitter in the nervous system. Within the 1C metabolic pathway choline generates methyl groups to reduce levels of homocysteine, this is done though betaine and betaine homocysteine methyltransferase. Our previous work has shown that when animals are deficient in dietary folic acid, choline is used [17].

3. I would like to suggest the authors to minimize their conclusions about the supposed inflammation processes, since they actually measured MMP-2, and -9 levels in fluids/tissues. The whole inflammatory scenario should be better described by analyses of many other biomarkers, such as cytokines (TNF-alpha, IL-10, IL-8, etc), as well as immune cell chemotactic compounds.

Response: We have added details of about the implications of our study to the conclusion section of the paper (page 9, lines 333 to 348). The revised text has been copied below for reference.

We report reduced levels of MMP-2 in brain and plasma tissue of male offspring 1.5 months after undergoing induced ischemic stroke from mothers deficient in choline. Previous research has shown that reduced levels of MMPs are associated with less severe injury and complications following cerebral ischemia. However, it would be an oversimplification to classify MMPs as inherently deleterious. MMPs are crucial players during tissue remodeling, repair, and healing processes, but their overexpression and uncontrolled activity can be detrimental, leading to excessive degradation of extracellular matrix components, contributing to inflammatory and pathological conditions. Further research is necessary to elucidate the precise benefits and adverse consequences of varying levels of MMPs in the brain of offspring from mothers with dietary deficiencies. Additionally, studying MMP expression at different time points post-stroke is crucial to understanding their dynamic roles and impacts on neurological recovery and damage. Furthermore, MMP-2 is constitutively expressed in several tissues and is regulated by tumor necrosis factor-α under the influence of NF-κB transcription factor [37] – investigating these levels in offspring brain tissue is also a future goal. As well as inhibiting MMP-2 function may provide some clues to the role of this inflammatory marker on offspring stroke outcome.

Reviewer 2 Report

Comments and Suggestions for Authors

1.       In the Abstract, the authors state: „We measured levels of matrix-metalloproteases (MMP)-2 and 9 in both plasma and brain tissue, and report reduced levels of MMP-2 in both, with no changes observed in MMP-9.“ It is not clear in which of the groups this was the case.

2.       The spaces after the full stops are not equal throughout the manuscript.

3.       The Introduction section is poorly written and should be proof read for academic English.

4.       The authors state „The aim of this study was to measure the levels of inflammatory markers MMP-2 and -9 post-stroke in the brain and plasma of female and male offspring from mothers deficient in dietary folic acid or choline during pregnancy and lactation.“. However, it is clear that they measure the markers in two intervention groups and compare it to the control group. This should be clearly stated as it is an important methodological point of the study. Furthermore, this should be clear throughout the paper.

5.       Why the data was presented as mean +/- standard error of the mean (SEM)? Please present it with standard deviation or median if more appropriate unless there is a specific reason not to do so (if yes, please explain).

6.       Line 190 „Four and half weeks after ischemic stroke“. Was this 31 or 32 days?

7.       In the first or second paragraph of the Materials and Methods section please state how many mothers were in each group and how many offspring were derived per group.

8.       The discussion should begin with a small summary of the main results

9.       The discussion is modes and above all lacks potential practical clinical implications.

10.   Please include the paragraph on limitations

11.   Conclusion should be based on the finding without discussion previous research

12.   Lines 324-327, please check the font

Comments on the Quality of English Language

The spaces after the full stops are not equal throughout the manuscript.

 The Introduction section is poorly written and should be proof read for academic English.

Author Response

  1. In the Abstract, the authors state: „We measured levels of matrix-metalloproteases (MMP)-2 and 9 in both plasma and brain tissue, and report reduced levels of MMP-2 in both, with no changes observed in MMP-9.“ It is not clear in which of the groups this was the case.

Response: We thank the reviewer for this comment. As requested, we have added the experimental group that had reduced levels of MMP-2 to the abstract on page 1 (lines 28 to 30). The revised text has been copied below for reference.

We measured levels of matrix-metalloproteases (MMP)-2 and 9 in both plasma and brain tissue, and report reduced levels of MMP-2 in ChDD male offspring in both tissue types.

  1. The spaces after the full stops are not equal throughout the manuscript.

Response: We apologize for this oversight and have gone through the manuscript in detail to remove any inconsistencies in formatting.

  1. The Introduction section is poorly written and should be proof read for academic English.

Response:  We apologize for this oversight and have revised the introduction, as well as proofread for English.

  1. The authors state „The aim of this study was to measure the levels of inflammatory markers MMP-2 and -9 post-stroke in the brain and plasma of female and male offspring from mothers deficient in dietary folic acid or choline during pregnancy and lactation.“. However, it is clear that they measure the markers in two intervention groups and compare it to the control group. This should be clearly stated as it is an important methodological point of the study. Furthermore, this should be clear throughout the paper.

Response: We thank the reviewer for this comment. We compared all three groups to each other, using two- and one-way ANOVAs (data analysis and statistics section, page 4 to 5, lines 196 to 205). We followed up any significant main effects with pairwise comparisons and reported both sets of results in the results section. We have clarified these comparisons within the data analysis and statistics section, results section and figure legends. The revised text has been copied below for reference.

2.6. Data Analysis and Statistics

GraphPad Prism 10.0 was used to analyze cell counts and ELISA data. In GraphPad Prism, D’Agostino-Pearson normality test was performed prior to two-way ANOVA analysis for all data. The two-way ANOVA compared the mean measurement of both sex and maternal dietary group for ELISA and immunofluorescence staining. One-way ANOVA analysis was used to compare maternal dietary differences when there were no offspring sex differences. Significant main effects of two- and one-way ANOVAs were followed up with Tukey’s post-hoc test to adjust for multiple comparisons. All data are presented as mean + standard error of the mean (SEM). Statistical tests were performed using a significant P value of 0.05.

Results section

Page 5, lines 212 to 213.

Male ChDD mice had lower levels of MMP-2 compared to FADD mice (p = 0.024)

Page 6, lines 219 to 211

Figure 2. Impact of maternal diet deficient in folic acid (FADD) or choline (ChDD) and control (CD) during pregnancy and lactation on levels of matrix-metalloproteases (MMP) -2 (A) and 9 (B) in plasma of female and male offspring 1 month after ischemic stroke. Mean + SEM of 10 to 14 animals per group. * p < 0.05, Tukey’s pairwise post-hoc analysis between male FADD and ChDD offspring.

Page 7, lines 239 to 246

Figure 3. Impact of maternal diets deficient in folic acid (FADD) or choline (ChDD) and control diet (CD) during pregnancy on lactation on neuronal levels of matrix-metalloproteases (MMP-2) in ischemic brain tissue of female and male offspring one month after ischemic stroke. Representative images of MMP-2 immunofluorescence colocalization with neuronal nuclei (NeuN) and 4',6-diamidino-2-phenylindole (DAPI) (A) and semi-quantitative quantification of MMP-2 levels within ischemic region of brain tissue. Mean + SEM of 3 to 4 animals per group. * p < 0.05, Tukey’s pairwise post-hoc analysis between male CD and ChDD offspring (one-way ANOVA analysis between males and significant maternal diet main effect).

Page 6, lines 234 to 236

Interestingly we see observed reduced levels of MMP-2 in male brain tissue because of maternal diet (F(2, 7) = 5.734, p = 0.034). Specifically, between ChDD and CD offspring (p = 0.023; Figure 3B).

5. Why the data was presented as mean +/- standard error of the mean (SEM)? Please present it with standard deviation or median if more appropriate unless there is a specific reason not to do so (if yes, please explain).

Response: Our previously published papers contain mean +/- standard error of the mean (SEM). We have consistently reported that data and it is important to us for this data reporting to be consistent across our different studies.

6. Line 190 „Four and half weeks after ischemic stroke“. Was this 31 or 32 days?

Response: For clarity we have revised the sentence (page 8, lines 315 to 316). The revised text has been copied below for reference.

From an experimental design perspective, these plasma and brain samples were taken 32 days (4.5 weeks after ischemic stroke) after ischemic stroke was induced.

  1. In the first or second paragraph of the Materials and Methods section please state how many mothers were in each group and how many offspring were derived per group.

Response: We have added the requested text to the end of paragraph 2 on page 3, lines 124 to 126. The revised text has been copied below for reference.

For each dietary group we had 6 female and male breeding pairs set up. The following number are the number offspring generated, CD (female: 15, male:13), ChDD (female: 13, male: 14), and FADD (female: 15, male: 12).

8. The discussion should begin with a small summary of the main results

Response: We thank the reviewer for this suggestion and have added a summary of the results to the beginning of the discussion (page 7, lines 262 to 265). The revised text has been copied below for reference.

This study aimed to investigate whether pro-inflammatory MMPs play a role in the impact of reduced maternal dietary levels of folic acid or choline on post-stroke outcomes in female and male offspring. We report that MMP-2 levels were decreased in the brain tissue and plasma of our choline deficient diet male offspring following ischemic stroke. Whereas MMP-9 levels were unaffected in treatment groups.

9. The discussion is modes and above all lacks potential practical clinical implications.

Response: We thank the reviewer for this comment, as it enhances our manuscript. Below are portions of the discussion that we think add some potential practical clinical implications.

Page 8, lines 294 to 298

Previous studies have shown that deficiencies in vitamin B (B6, B12, & folate) can elevate homocysteine levels, which in turn can increase MMP activity, contributing to neurovascular and brain damage [13,35]. Deficiencies in vitamin E and other antioxidants like selenium can also results in oxidative stress, increasing MMP activity in the brain [36,37].

Page 9, lines 342 to 348

Additionally, studying MMP expression at different time points post-stroke is crucial to understanding their dynamic roles and impacts on neurological recovery and damage. Furthermore, MMP-2 is constitutively expressed in several tissues and is regulated by tumor necrosis factor-α under the influence of NF-κB transcription factor [38] – investigating these levels in offspring brain tissue is also a future goal. As well as inhibiting MMP-2 function may provide some clues to the role of this inflammatory marker on offspring stroke outcome.

  1. Please include the paragraph on limitations

Response: We have added limitations of our study to the discussion, the additional text is on pages 8 and 9, lines 317 to 330. The revised text has been copied below for reference.

From an experimental design perspective, these plasma and brain samples were taken 32 days (4.5 weeks after ischemic stroke) after ischemic stroke was induced. Results could have potentially differed if samples were taken at different points in time, such as sample collection right after ischemic stroke. Potential future studies could include analyzing the same samples across different time frames and assessing if different levels of MMPs are noticed at different points in time. Previous research measured MMP levels at different times following cardiac ischemia. They noticed that MMP levels are increased in proportion to level of ischemia at the initial onset of ischemia (acute myocardial infarction vs stable coronary artery disease), but later decrease to below less severe ischemic levels [39]. This could potentially explain the results of our study. The diet deficient treatment groups may have undergone more severe ischemia than the control groups, having more MMPs initially but could potentially still have lower levels of MMP a short time later. The previously mentioned study regarding heart ischemia took their samples 3 months post ischemic injury, our results show similar findings at the 1.5-month mark.

11. Conclusion should be based on the finding without discussion previous research

Response: We have revised out conclusion to not include a discussion of previous research, this is completed on page 9, lines 331 to 346. The revised text has been copied below for reference.

We report reduced levels of MMP-2 in brain and plasma tissue of male offspring 1.5 months after undergoing induced ischemic stroke from mothers deficient in choline. Previous research has shown that reduced levels of MMPs are associated with less severe injury and complications following cerebral ischemia. However, it would be an oversimplification to classify MMPs as inherently deleterious. MMPs are crucial players during tissue remodeling, repair, and healing processes, but their overexpression and uncontrolled activity can be detrimental, leading to excessive degradation of extracellular matrix components, contributing to inflammatory and pathological conditions. Further research is necessary to elucidate the precise benefits and adverse consequences of varying levels of MMPs in the brain of offspring from mothers with dietary deficiencies. Additionally, studying MMP expression at different time points post-stroke is crucial to understanding their dynamic roles and impacts on neurological recovery and damage. Furthermore, MMP-2 is constitutively expressed in several tissues and is regulated by tumor necrosis factor-α under the influence of NF-κB transcription factor [37] – investigating these levels in offspring brain tissue is also a future goal. As well as inhibiting MMP-2 function may provide some clues to the role of this inflammatory marker on offspring stroke outcome.

  1. Lines 324-327, please check the font

Response: Thank-you for bringing this information to our attention, we have revised the font (page 8, lines 324 to 336). The revised text has been copied below for reference.

However, it would be an oversimplification to classify MMPs as inherently deleterious. MMPs are crucial players during tissue remodeling, repair, and healing processes, but their overexpression and uncontrolled activity can be detrimental, leading to excessive degradation of extracellular matrix components, contributing to inflammatory and pathological conditions. Further research is necessary to elucidate the precise benefits and adverse consequences of varying levels of MMPs in the brain of offspring from mothers with dietary deficiencies. Additionally, studying MMP expression at different time points post-stroke is crucial to understanding their dynamic roles and impacts on neurological recovery and damage. Furthermore, MMP-2 is constitutively expressed in several tissues and is regulated by tumor necrosis factor-α under the influence of NF-κB transcription factor [50] – investigating these levels in offspring brain tissue is also a future goal. As well as inhibiting MMP-2 function may provide some clues to the role of this inflammatory marker on offspring stroke outcome.

Round 2

Reviewer 2 Report

Comments and Suggestions for Authors

I have no further comments